# Physical Activity and Sport for Acquired Brain Injury (PASABI): A Non-Randomized Controlled Trial

**DOI:** 10.3390/medicina57020122

**Published:** 2021-01-29

**Authors:** Marta Pérez-Rodríguez, Saleky García-Gómez, Javier Coterón, Juan José García-Hernández, Javier Pérez-Tejero

**Affiliations:** 1“Fundación Sanitas” Chair in Inclusive Sport Studies (CEDI), Department of Health and Human Performance, Faculty of Physical Activity and Sport Sciences (INEF), Universidad Politécnica de Madrid, 28040 Madrid, Spain; saleky@gmail.com (S.G.-G.); j.perez@upm.es (J.P.-T.); 2AFIPE Research Group, Universidad Politécnica de Madrid, 28040 Madrid, Spain; j.coteron@upm.es; 3Department of Social Sciences Applied to Sport, Physical Activity and Leisure, Faculty of Physical Activity and Sport Sciences (INEF), Universidad Politécnica de Madrid, 28040 Madrid, Spain; 4“Sport for ABI Club”, Francisco de Vitoria University, 28223 Madrid, Spain; juanjo@deporteparadca.com

**Keywords:** physical activity, health-related quality of life, acquired brain injury, chronic phase, sport

## Abstract

*Background and objectives:* Acquired brain injury (ABI) is the first cause of disability and physical activity (PA) is a key element in functional recovery and health-related quality of life (HRQoL) during the subacute and chronic phases. However, it is necessary to develop PA programs that respond to the heterogeneity and needs of this population. The aim of this study was to assess the effectiveness of a PA program on the HRQoL in this population. *Materials and Methods:* With regard to recruitment, after baseline evaluations, participants were assigned to either the intervention group (IG, n = 38) or the control group (CG, n = 35). Functional capacity, mood, quality of life and depression were measured pre- and post-intervention. The IG underwent the “Physical Activity and Sport for Acquired Brain Injury” (PASABI) program, which was designed to improve HRQoL (1-h sessions, two to four sessions/week for 18 weeks). The CG underwent a standard rehabilitation program without PA. *Results:* Results for the IG indicated significant differences and large effect sizes for the physical and mental dimensions of quality of life, as well as mood and functional capacity, indicating an increase in HRQoL. No significant differences were found for the CG across any variables. *Conclusions:* The PASABI program was feasible and beneficial for improving physiological and functionality variables in the IG. The wide range of the activities of the PASABI program allow its application to a large number of people with ABI, promoting health through PA, especially in the chronic phase.

## 1. Introduction

Acquired brain injury (ABI) requires specific training programs based in neurorehabilitation with the aim of achieving the highest possible level of autonomy and participation during the subacute and chronic phases [1]. The consequences of an ABI are different in each person, and it is usual to find deficits in the physical, cognitive, behavioral, and sensorial dimensions. As a result, a multidisciplinary approach is fundamental for the treatment of ABI [2]. The incidence of ABI increases each year, with the two most common etiologies being traumatic brain injury (TBI) and stroke [3,4].

Health-related quality of life (HRQoL) includes the physical, psychological, and social dimensions, which requires the evaluation of different variables to know this multidimensional concept [5]. Literature points out that HRQoL is lower in people with an ABI compared to that in people without an ABI due to the sedentary lifestyle, generating secondary problems [6,7], and social leisure activities are recommended to promote the highest HRQoL [6]. In this line, physical activity (PA) has demonstrated being an important tool to improve not only physical affectation [8,9] but also cognitive [10,11] and behavioral sequels [12], being a key to socialize, to increase reintegration into the community, and improve HRQoL [13]. It seems in the literature that there is no consensus on the type of PA program that should be proposed to people with chronic ABI, whose main goal should be to improve HRQoL, coinciding with interest in group activities [14,15].

The International Classification of Functioning (ICF) developed by the World Health Organization (WHO) [16] is the bio-psycho-social framework to design and implement PA programs for people with disability and specifically used for functional evaluation and PA participation in people with ABI [17]. For all of the above mentioned, this study aimed to assess the efficacy of the “Physical Activity and Sport Brain Injury” (PASABI) program in the HRQoL of people with ABI during the subacute and chronic phases. It was hypothesized that involvement in the PASABI program would improve mood, depression, quality of life, functional capacity, and participation.

As the systematic review carried out by Jones et al. [18] points out, intervention programs are generally not described in depth, thus preventing their reproducibility. Most of the research includes specific proposals for stroke or TBI, although people with ABI in rehabilitation centers receive the same therapies. The two most practiced PA programs in the literature are the treadmill and exercise bike, which are not group activities [14,15].

## 2. Materials and Methods

### 2.1. Participants

A total of 73 subjects were recruited into this study according to the following inclusion criteria: (a) being over 18 years of age; (b) not having language problems that make oral and written comprehension impossible; (c) being able to walk independently or with auxiliary material; (d) having had at least 1 year of evolution since the ABI; (e) having received at least 3 months of rehabilitation for the ABI; (f) residing at home and not in a rehabilitation center; (g) not having resumed working life; and (h) having a sufficient level of cognitive functioning to answer questionnaires coherently. 

With regard to recruitment (Figure 1), after baseline evaluations, participants were assigned, using a mixed paired design, to either the intervention group (IG) or the control group (CG). The assignment of each group was done according to the preference of the participants for ethics and feasibility conditions. 

### 2.2. Study Design

The present study was a non-randomized clinical trial. Pre-post intervention design, with measurements at baseline and after 18 weeks of intervention. 

### 2.3. Outcome Measures

Different instruments, according to the dimensions of the ICF, were used to assess the program’s effects on the HRQoL of participants. Table 1 shows the variables of study of HRQoL that have been established. The measurements were made at baseline and after 18 weeks of intervention in both the IG and CG. 

For demographic data, a collection form was used with the sociodemographic data referring to date of birth, gender, housing, profession prior to the ABI, etiology of the injury, months since the injury, type of displacement, mobility, prior PA practice, and whether the participant was currently receiving therapies in a rehabilitation center or not. Additionally, the Spanish version of the Beck Depression Inventory II (BDI-II) (Sanz and Vázquez, 1998) was used to evaluate depressive symptoms. Mood with respect to factors of a negative (anger, fatigue, tension, and depressed state) and positive (vigor and friendliness) nature were evaluated using the Spanish version of the Profile of Mood States (POMS) [28]. 

Furthermore, the SF-36 version 1.4 was used to assess quality of life [29,30]. This tool evaluates the self-perception of health and quality of life. In addition, the level of activity was measured through the Global Physical Activity Questionnaire (GPAQ) [31]. This questionnaire contains the dimensions work, travel, and free time. Finally, the 6 Minute Walk Test (6MWT) was selected to assess functional capacity [32].

### 2.4. Intervention

First, participants were provided with general information about the study and informed consent, in addition to recording demographic data. Then, general cognitive functioning, depressive symptoms, mood, quality of life, activity level, and functional ability were assessed before and after 18 weeks of the PASABI program. The 18-week exercise program was developed in six municipal sports centers of Madrid in people with ABI in the chronic phase, with participants performing two to four sessions per week at 1 h per session. 

#### Physical Activity, Sport and Acquired Bran Injury (PASABI)

The PASABI program aims to increase autonomy in the conduct of DLAs, generate social relationships, and achieve greater reintegration into the community. It defines the type of activity to be carried out related to the level of affectation and proposes specific tasks focused on key aspects of improving the HRQoL of people with ABI. It includes games as a key element, because they are a resource that involves fun, socialization, interactions that incite instinctive learning, organization, and the development of complex skills not only for the child but also for the adult. Furthermore, diverse activities designed according to abilities based on the proposal of García-Hernández and Pérez-Rodríguez [33], as well as previous studies that established the benefits of PA on the health of people with ABI [18,33,34,35]. The activities of the PASABI program include water activity, swimming, paddle tennis, initiation to football and initiation to athletics. Within the characteristics of the structure of the session. The proposed PASABI program included a first phase with a warm-up exercise (5–10 min of joint mobility tasks) and a second phase with at least three specific analytical exercises and three specific AFA exercises. Moreover, an interactive part with at least one game. Finally, return to the calm with 5 min of breathing control tasks and 10 min stretching the muscle groups involved. Two adapted PA professionals supervised the intervention.

### 2.5. Statistical Data

For descriptive statistics, data were checked for normality using the Kolmogorov–Smirnov test, which indicated the need for non-parametric tests. A descriptive analysis (by group) of the variables studied was carried out using, as reference values, the means and standard deviations for continuous variables and frequencies and percentages for categorical variables. The chi-square test was used for categorical variables, and the Mann–Whitney U test was used for continuous variables. After the intervention, the Wilcoxon W test of related samples was used to establish intragroup differences, and the Mann–Whitney U test of independent samples was used to analyse the differences between groups. Cliff’s Delta was calculated for the estimation of the non-parametric effect size in the comparisons. An r-value higher than 0.1, 0.3 or 0.5 was considered as a small, medium, or large effect size, respectively [36]. The significance level was set at *p* ≤ 0.05, and the SPSS 22.0 statistical package was used. 

### 2.6. Ethics

All participants signed the informed consent form. The design of the research was supervised and approved by the Ethics Committee of the leading institution of the study, and the recommendations of the Declaration of Helsinki were followed at all times [37]. This protocol is registered in Clinical Trial.gov (NCT03162484).

## 3. Results

Demographic characteristics of all the participants in the IG and CG at baseline are presented in Table 2. Of the 38 people with ABI assigned to the IG, only 34 were included in the intervention. One participant was lost to follow-up. Additionally, three participants did not complete the 18-week intervention for personal reasons. At 18 weeks, more than 80% of this sample showed adherence to the program. On other hand, in the CG, seven participants were included—four participants did not pass the neuropsychological assessment and three participants did not complete the planning program.

Table 3 shows the results for each variable at baseline and after 18 weeks of intervention according to group. After the intervention, significant differences were found in the IG for the variables sedentary, functional capacity, physical function (*p* < 0.001), and mental health (*p* < 0.05), producing an increase in the scores, as well as a generalized decrease in the scores of the variables, depression-B, depression, anger (*p* < 0.05), and stress (*p* < 0.01). Regarding the QA score, significant differences were found in the variables sedentary (*p* < 0.01), friendship and fatigue (*p* < 0.05), producing a general increase in the scores, also in the IG. 

With respect to the comparison of scores after the intervention, differences were found between the QA and IG in the variables sedentary (U = 359.50, *p* < 0.05, r = 0.28), functional capacity (U = 360.00, *p* < 0.05, r = 0.28), physical function (U = 402.00, *p* < 0.05, r = 0.32), showing the CG with the highest score in the sedentary variable, and the IG in the functional capacity and physical function variables.

## 4. Discussion

This study assessed the efficacy of a group PA program for people with ABI in the chronic phase, proposing a new approach and a line of work to be explored in the future for this population. The PASABI program has proven to be feasible. Furthermore, the adherence to the program was very high. Out of 39 participants, none of them left the program, and all exceeded 80% attendance at the sessions. In our opinion, the design based on the ICF framework succeeds in not only responding to the functional characteristics but also contextual factors near their place of residence and minimizing barriers to practice [38]. This along with the professionalism of the staff team in the referral of each participant to the most suitable activity was critical to the success of the program in terms of participation and effectiveness [33].

No adverse events occurred during the sessions, and no participants reported secondary problems or complications arising from practice. The length of the intervention with regard to the session time (1 h) and the 18 weeks of intervention could have been decisive in the positive results of the study. Likewise, a frequency of two to four sessions per week allowed participants to adapt to their possibilities and led to a transfer beyond the study. Most participants in the IG continued to engage in the program after the completion of the study.

Previous studies considered age as a predictor of PA practice; however, the mean age between the two groups evaluated was similar, and the IG had greater age differences between the minimum and the maximum. Likewise, the results revealed that the severity of the injury and the sport practice before the injury could be indicators for participation after ABI in PA programs [39].

With respect the content of the PASABI program, it seems that group activities like those provided by the program are efficient and allow participation and high rates of engagement, as mentioned. Regarding aquatic activities, we agree with authors in affirming the effectiveness of this type of exercise to improve HRQoL [34,40].

In relation to the improvement of the quality of life measured with the SF-36, the results indicated significant improvements of the IG in the physical and mental dimensions, which corroborates the capacity of a group PA to generate benefits in the different areas, coinciding with authors who also point this out [41,42]. The results on mood indicate a significant improvement in the dimensions of depression, cholera, and tension, coinciding with expert authors on this topic [25,43]. With respect to fatigue, the dimensions included in the POMS increased significantly in the CG after the 18-week intervention period. This could be justified by the low physical condition of the CG that leads to lower physical levels and a greater sense of fatigue at any effort.

Due to the importance of depression in the functional recovery of people with ABI, in this study, the BDI-II was used to assess depressive symptoms. A tendency to decrease in the IG and increase in the CG after the intervention was observed, coinciding with other authors who found improvement in depressive symptoms after an intervention [12,44] and stroke. 

Moreover, the findings show that there was no improvement in the level of activity after the intervention. It should be noted that the increase in the score in the sedentary variable was greater in the CG, although it increased in both groups. This data may be due to the fact that 56% of the IG did not receive therapy in rehabilitation centers or return to work, compared to 100% of the CG, who did receive therapy during this time, providing more active hours. Nonetheless, it is remarkable that the IG has increased physical function and functional capacity; although, the hours of activity did not increase. So, in order to achieve an adequate physical condition and an adequate mood, it seems essential to perform PA as an indispensable complement to the standard rehabilitation programs in the chronic phase [45].

All studies confirm the impact of functional capacity on HRQoL in people with ABI [23]. Studies show improvement in functional capacity after a PA program, either individual or group from a minimum of eight weeks [9,46]. The 6MWT appears to be a reliable and valid indicator of functional capacity [47,48].

As limitations of this research, we can indicate that the study sample was limited to voluntarily participating individuals, leading to a possible selection bias. The sample consisted of participants mildly to moderately affected by ABI, not including people with more affectation given the characteristics of the PASABI program. The positive results of the study in a population with ABI open a fruitful line of research in the role of PA-oriented health in this population, with application to other ABI levels of impairment. In this regard, this study could be a starting point for future investigations with a randomized clinical trial design. 

## 5. Conclusions

The bio-psycho-social framework proposed by the ICF is adequate to encompass all aspects to be taken into account in the design of a group PA program in people with chronic ABI.

Having professional staff, minimizing barriers, and selecting schedules and facilities close to the place of residence are critical to achieving a high level of attendance and adherence to the program.

This research has shown that the PASABI program, with content based on different sports modalities and adapted activities designed specifically for people with ABI in the chronic phase to achieve the highest autonomy and participation, has positive effects on HRQoL.

## Figures and Tables

**Figure 1 medicina-57-00122-f001:**
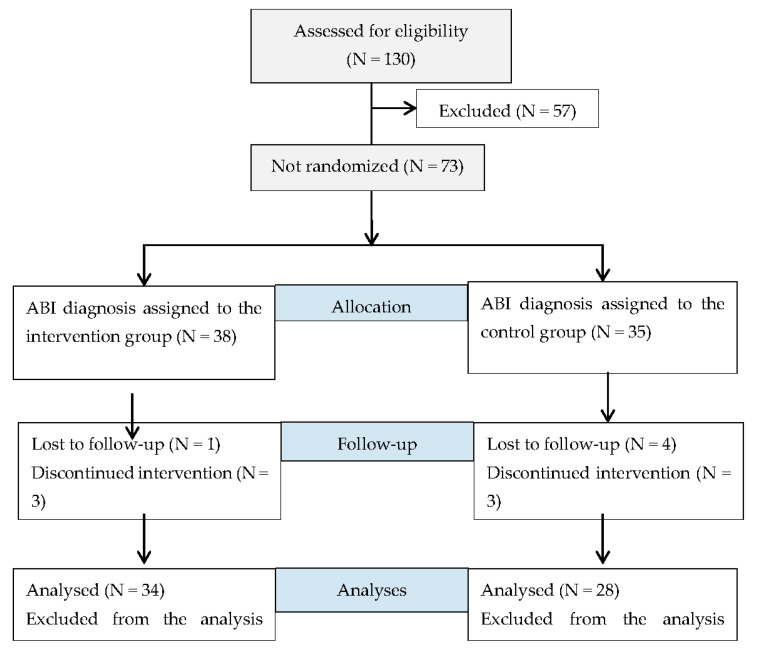
Consort flow diagram for participants. (Acquired brain injury—ABI).

**Table 1 medicina-57-00122-t001:** Study variables of health-related quality of life (HRQoL) based on International Classification of Functioning (ICF) domains.

ICF	Variable	Description	Instrument	Author
	Functional capacity	Gait	6 min Walk Test	[19,20,21,22]
Body functions and structures	Quality of life	Quality of life	SF-36	[23]
	Psychological condition	Mood	Profile of Mood States (POMS)	[24,25]
		Depression	Beck’s Inventory	[10,22,26]
Activities and participation	Participation	Getting involved in the task	GPAQ	[21,27]
Contextual factor	Personal and environmental factors	Sociodemographic data	Registration forms	

**Table 2 medicina-57-00122-t002:** Baseline demographic characteristics of participants.

	Intervention Group (n = 34)	Control Group (n = 28)	*p*
Gender			*p* < 0.05
Male	22 (64.7%)	13 (46.4%)	
Female	12 (35.3%)	15 (53.6%)	
Age (years)	49.53 ± 16.1	51.5 ± 8.4	*p* ≥ 0.05
Lesion etiology			*p* < 0.001
Stroke	22 (64.7%)	19 (67.9%)	
Head trauma	8 (23.5%)	4 (14.3%)	
Encephalopathy	0	3 (10.7%)	
Brain tumor	4 (11.8%)	2 (7.1%)	
Displacement			*p* < 0.001
Dependent autonomy	11 (32.4%)	14 (50%)	
Independent autonomy	23 (67.6)	13 (46.4%)	
Wheelchair use	0	1 (3.6%)	
Mobility			*p* < 0.05
Public transport	15 (44.1%)	6 (21.4%)	
Own car	7 (20.6%)	3 (10.7%)	
Public transport with family	3 (8.8%)	5 (17.9%)	
Car with family	9 (26.5%)	14 (50%)	
Housing			*p* < 0.001
Lives alone	4 (11.8%)	3 (10.7%)	
With couples	17 (50%)	17 (60.7%)	
With parents	11 (32.4%)	4 (14.3%)	
With children	2 (5.9%)	2 (5.9%)	

Note: Values are mean (SD) for continuous variables and frequency (%) for categorical variables. The chi-square test was used for categorical variables, and the Mann–Whitney U test was used for continuous variables.

**Table 3 medicina-57-00122-t003:** SF-36, POMS, GPAQ, BDI-II, and 6MWT at baseline and post-intervention assessments.

			Intervention Group (n = 39)			Control Group (n = 28)			Comparison after Intervention	
			**X (SD)**	**Z**	**r**	**X (SD)**	**Z**	**r**	**Z**	**r**
SF-36	Physical Function	Pre	66.03 (17.66)	−2.60	0.33	57.5 (27.64)	−0.79	0.10	−2.51	0.32
		Post	72.65 (19.32)			58.93 (21.7)			−0.86	0.11
	Physical role	Pre	73.53 (34.23)	−0.08	0.01	75.89 (34.35)	−1.31	0.17	−0.37	0.05
		Post	73.53 (30.74)			65.18 (36.22)			−0.73	0.09
	Pain	Pre	69.03 (26.38)	−0.46	0.06	66.71 (23.89)	−0.16	0.02	−0.83	0.11
		Post	71.32 (23.84)			67.43 (27.42)			−0.20	0.03
	General health	Pre	61.41 (16.21)	−1.81	0.23	61.21 (20.72)	−0.74	0.09	−1.42	0.18
		Post	66.85 (20.15)			63.07 (21.92)			−1.21	0.15
	Vitality	Pre	62.06 (18.47)	−0.79	0.10	64.29 (20.89)	−1.47	0.19	−0.42	0.05
		Post	63.82 (18.91)			59.82 (19.22)			−0.78	0.10
	Social Function	Pre	70.59 (21.95)	−1.93	0.25	75 (25.23)	−1.13	0.14	−1.49	0.19
		Post	79.04 (19.39)			78.57 (18.28)			−1.18	0.15
	Emotional Role	Pre	79.41 (37.62)	−0.32	0.04	65.48 (44.89)	−1.09	0.14	−1.52	0.19
		Post	82.35 (35.04)			72.62 (38.55)			−1.36	0.17
	Mental Health	Pre	66.59 (17.99)	−2.1	0.27	70.14 (15.72)	−0.88	0.11	−0.80	0.10
		Post	73.76 (15.13)			68 (19.26)			−2.24	0.28
	Health Transition	Pre	2.15 (0.86)	−0.83	0.11	2.11 (1.03)	−0.74	0.09	−0.84	0.11
		Post	2.03 (0.76)			2.21 (1.1)			−2.19	0.28
POMS	Vigor	Pre	11.85 (4.33)	−1.45	0.18	10.43 (5.6)	−1.70	0.22	−2.51	0.32
		Post	12.76 (4.18)			11.39 (5.2)			−0.86	0.11
	Friendship	Pre	13.91 (3.25)	−0.46	0.06	13.86 (3.85)	−2.01	0.26	−0.37	0.05
		Post	14.24 (3.62)			15.36 (3.64)			−0.73	0.09
	Fatigue	Pre	5.53 (3.96)	−0.04	0.00	5.61 (5.51)	−2.19	0.28	−0.83	0.11
		Post	5.5 (3.92)			7.5 (5.6)			−0.20	0.03
	Depression	Pre	4.79 (3.81)	−2.14	0.27	4.46 (4.48)	−1.10	0.14	−1.42	0.18
		Post	3.65 (4.1)			5.5 (4.87)			−1.21	0.15
	Cholera	Pre	4.15 (3.36)	−2.39	0.30	5.18 (4.41)	−0.92	0.12	−0.42	0.05
		Post	2.76 (2.48)			4.54 (4.29)			−0.78	0.10
	Tension	Pre	5.74 (4.12)	−2.67	0.34	5.61 (5.1)	−0.39	0.05	−1.49	0.19
		Post	3.59 (3.09)			5.36 (5.72)			−1.18	0.15
GPAQ	Sedentary lifestyle	Pre	461.1(249.09)	−2.98	0.38	576.4 (283.36)	−2.60	0.33	−1.52	0.19
		Post	651.4 (186.25)			755.4 (179.31)			−1.36	0.17
Beck	Depression	Pre	11.71 (8.35)	−2.06	0.26	10.71 (10.36)	−0.78	0.10	−0.80	0.10
		Post	9.53 (7.89)			11.71 (9.15)			−2.24	0.28
6MWT	Functional capacity	Pre	355.7 (139.78)	−2.88	0.37	295.7 (134.9)	−0.65	0.08	−0.84	0.11
		Post	396.5 (158.45)			305.4 (153.52)			−2.19	0.28

Note: Values X = mean (SD = standard deviation). *p*-Values for the change scores between the two groups were obtained using the Mann–Whitney U test. Within-group p was obtained using the Wilcoxon test. r values with Cliff’s Delta. Level of significance *p* < 0.05. (Short form 36 questionnaire—SF-36, Profile of Mood States—POMS, Global Physical Activity Questionnaire—GPAQ, Beck Depression Inventory-II—BDI-II), and Six Minutes Walk Test—6MWT).

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
