# Peer review of "Physical Activity and Sport for Acquired Brain Injury (PASABI): A Non-Randomized Controlled Trial"

_medicina, 2021, doi:10.3390/medicina57020122_

Round 1

Reviewer 1 Report

Authors have done quite a few changes in the manuscript including grammatical changes. It does make the manuscript more clearer.

Reviewer 2 Report

The paper should now be published.

This manuscript is a resubmission of an earlier submission. The following is a list of the peer review reports and author responses from that submission.

Round 1

Reviewer 1 Report

It is an interesting article. Why randomization was not done for the subjects? Considering some questionnaires are subjective, it is highly likely that subjects and investigators would be biased. Are we sure about reliability of the data?

Reviewer 2 Report

The study presents a clinical trial related to the validation of PASABI method (Physical activity and Sport for Acquired Brain Injury). The presentation is very good and clear. The authors propose two groups: an intervention group and a control group, with people which had brain injures. The intervention group follows the PASABI method (which contains also group exercises) and the control group follows the tradition method of recovery (bike, treadmill exercises). PASABI proved to be very efficient, having a positive impact on Health Related Quality of Life (HRQoL). The authors should check again the numbers from the study: "PASABI program has proven to be feasible and the adherence to the 199 program was very high, as from the 39 participants, none of them left the program". In the Figure 1. Consort flow diagram for participants I had the impression that other numbers were given. 

Reviewer 3 Report

this paper is very interesting subject evaluated by a good methodology which is not easy in these domains with multifactorial treatments and heterogenous populations